

# Evaluating Snow Microwave Radiative Transfer (SMRT) model emissivities using observations of Arctic tundra snow

Kirsty Wivell[1], Stuart Fox[1], Melody Sandells[2], Chawn Harlow[1], Richard Essery[3], and Nick Rutter[2]

[1]Met Office, Fitzroy Road, Exeter, UK
[2]Department of Geography and Environmental Sciences, Northumbria University, Newcastle upon Tyne, UK
[3]Department of Geosciences, University of Edinburgh, Edinburgh, UK

**Correspondence:** Kirsty Wivell (kirsty.wivell@metoffice.gov.uk)

**Abstract.** Improved modelling of snow emissivity is needed to improve the assimilation of surface-sensitive atmospheric sounding observations from satellites in polar regions for Numerical Weather Prediction (NWP). This paper evaluates emissivity simulated with the Snow Microwave Radiative Transfer (SMRT) model using observations of Arctic tundra snow, at frequencies between 89 and 243 GHz. Measurements of snow correlation length, density and layer thickness were used as

input to SMRT, and an optimization routine was used to assess the impact of each parameter on simulations of emissivity when compared to a set of Lambertian emissivity spectra, retrieved from observations of tundra snow from three flights of the Facility for Airborne Atmospheric Measurements (FAAM) aircraft. Probability distributions returned by the optimization routine demonstrate parameter uncertainties and the sensitivity of simulations to the different snow parameters. Results showed that SMRT was capable of reproducing a range of observed emissivities between 89 and 243 GHz. Varying correlation length alone

allowed SMRT to capture much of the variability in the emissivity spectra, however, overall RMSE (and MAE) decreased from 0.029 (0.018) to 0.013 (0.0078) when the thickness of the snow layers was also varied. When all three parameters were varied simulations were similarly sensitive to both correlation length and density, although the influence of density was most evident when comparing spectra from snowpacks with and without surface snow. Simulations were most sensitive to surface snow and wind slab parameters, while sensitivity to depth hoar depended on the thickness and scattering strength of layers above,

demonstrating the importance of representing all three parameters for multi-layer snowpacks when modelling emissivity spectra. This work demonstrates the ability of SMRT to simulate snow emissivity at these frequencies, and is a key step in progress towards modelling emissivity for data assimilation in NWP.



# 1 Introduction

Satellite microwave sounding observations in the Arctic enhance the skill of Numerical Weather Prediction (NWP) forecasts for both high and mid-latitude regions (Lawrence et al., 2019; Duncan et al., 2021; Laroche and Poan, 2021). The high spatial
and temporal coverage of polar orbiting satellites makes them an important source of sounding data over the high latitudes where conventional observations are limited. Instruments such as the Microwave Humidity Sounder (MHS), the Advanced Microwave Sounding Unit A (AMSU-A), and the Advanced Technology Microwave Sounder (ATMS), and up-coming instruments, the Microwave Sounder (MWS) and Microwave Imager (MWI), due to be launched on Metop Second Generation (Metop-SG) satellites in the mid 2020s (EUMETSAT), measure key atmospheric sounding channels. Currently however, many
surface-sensitive observations, i.e. channels with weighting functions that peak in the lower troposphere, are not assimilated in polar regions, particularly in winter, due to the highly variable surface emissivity of snow and sea ice (Geer et al., 2014). Snow emissivity depends on snowpack microstructure and stratigraphy, which evolve over time due to precipitation, wind redistribution, and grain metamorphosis (Grody, 2008). This seasonal evolution introduces uncertainty into emissivity retrievals and modelling, particularly given the sparse in situ snow data in these regions and the influence of variability in snowpack layering
and microstructure on scattering.

Accurate modelling of emission from snow covered surfaces could improve the assimilation of surface affected radiances over current methods where NWP systems rely on emissivity estimates from brightness temperature ($T_b$) measurements in window channels, or from emissivity atlases (Geer et al., 2014; Lawrence et al., 2019; Hirahara et al., 2020). To achieve this, NWP models require both accurate predictions of the physical snow properties which impact surface emission for layered
snowpacks, and a reliable snow microwave emissivity model driven by these physical properties, to be used by the radiative transfer forward operator in an assimilation system. Improvements have been made to the physical modelling of snow in land surface models, such as the introduction of multi-layer snow schemes in the Met Office JULES (Joint UK Land Environment Simulator; Walters et al., 2019) and ECMWF H-TESSEL (Hydrology-Tiled ECMWF Scheme for Surface Exchanges over Land; Arduini et al., 2019) models. Further improvements can be made through better understanding and representation of the
snow parameters which are most critical to microwave emissivity, and the processes that influence them.

This paper focuses on the microwave emissivity modelling aspect of microwave radiance assimilation for NWP. Previous studies (Harlow and Essery, 2012; Hirahara et al., 2020) have assessed emissivity simulations from snow microwave emission models such as the Microwave Emission Model of Layered Snowpacks (MEMLS) and the multi-layer Helsinki University of Technology (HUT) snow model. Hirahara et al. (2020) demonstrated the positive impact of accurate microwave emissiv-
50 ity modelling in an integrated forecasting system using the ECMWF Community Microwave Emission Modelling platform (CMEM), which contains the multi-layer HUT snow model. However, improvements were only seen up to 20 GHz, highlighting the need for improved emissivity modelling at higher frequencies. Harlow and Essery (2012) used MEMLS to simulate emissivity spectra between 89 and 183 GHz, which were compared to area-averaged emissivity spectra observed from an aircraft. They found that MEMLS simulations were most sensitive to parameters such as correlation length and thickness in the
55 top most layers of the snow, but found that MEMLS could not simulate all emissivity spectra within measurement uncertainty





without accounting for surface roughness or Mie scattering. Harlow and Essery (2012) were limited by a lack of quantitative measurements of microstructure, therefore snow type profiles measured in the field were related to correlation length bounds from the literature (Wiesmann et al., 1998).

The Snow Microwave Radiative Transfer (SMRT; Picard et al., 2018) model has been developed as a modular active/passive
microwave radiative transfer model for multi-layer snow, offering flexibility in the choice of electromagnetic and microstructure models. This paper aims to assess SMRT's ability to simulate snow microwave emissivities between 89 and 243 GHz. The MACSSIMIZE (Measurements of Arctic Clouds, Snow and Sea Ice nearby the Marginal Ice ZonE) field campaign in Trail Valley Creek (TVC), Canada in 2018, provided a dataset of physical snow properties observed across 29 snow pits, as well as ground and airborne microwave $T_b$ measurements. The snow pit observations include quantitative measurements of microstruc-
ture in the form of specific surface area (SSA), which improves on visual grain size estimate datasets from previous campaigns. These observations were of tundra snow, which covers a large proportion of the Arctic, and is associated with relatively shallow snowpacks composed of large grained low density depth hoar, overlain by fine grained high density wind slab and low density fresh snow layers (Sturm et al., 1995). SMRT simulations of $T_b$ between 89 and 243 GHz for three-layer snowpacks, driven by observed snow properties from MACSSIMIZE, were evaluated in Sandells et al. (2023). Although simulations gave reasonable
agreement with ground and airborne observations at 89 GHz, there was less agreement at higher frequencies, with snow pit simulations and airborne observations having statistically different distributions, even when accounting for the atmospheric contribution. A key challenge in the analysis was the spatial disparity between point-based simulations and the larger spatial scale of airborne observations.

This paper attempts to address the limitations of Sandells et al., by using a Markov Chain Monte Carlo (MCMC) algorithm
to randomly search over the full range of TVC snow pit observations for sets of parameter values that, when used as input to SMRT, produce simulated emissivities that match a set of observed emissivity spectra. To do this, SMRT was coupled to the Shuffled Complex Evolution Metropolis (SCEM-UA; Vrugt et al., 2003) algorithm, based on the method of Harlow and Essery (2012). Observed spectra between 89 and 243 GHz were generated by clustering approximately 2000 airborne observations from MACSSIMIZE, and were deemed to be representative of variability in tundra snow emissivity. Results
provide a probability distribution for parameters of interest, which indicates the sensitivity of emissivity simulations to different snow parameters. The key aims of this paper are to assess the ability of SMRT to simulate a range of emissivity spectra within the constraints of observed snow parameters, and to identify which parameters are most important for accurate modelling of surface emissivity, as these parameters will need to be represented for emissivity modelling to be effective in an assimilation system.

The paper is structured as follows: Sect. 2 introduces the MACSSIMIZE campaign (2.1), the airborne data and method for retrieving observed emissivities (2.2) and ground-based data collected during the campaign (2.3); followed by descriptions of SMRT (2.4), the MCMC sampler SCEM-UA (2.5), and the setup of the retrievals (2.6). Section 3 presents the results of the MCMC retrievals, and discussion and conclusions are given in Sect. 4.



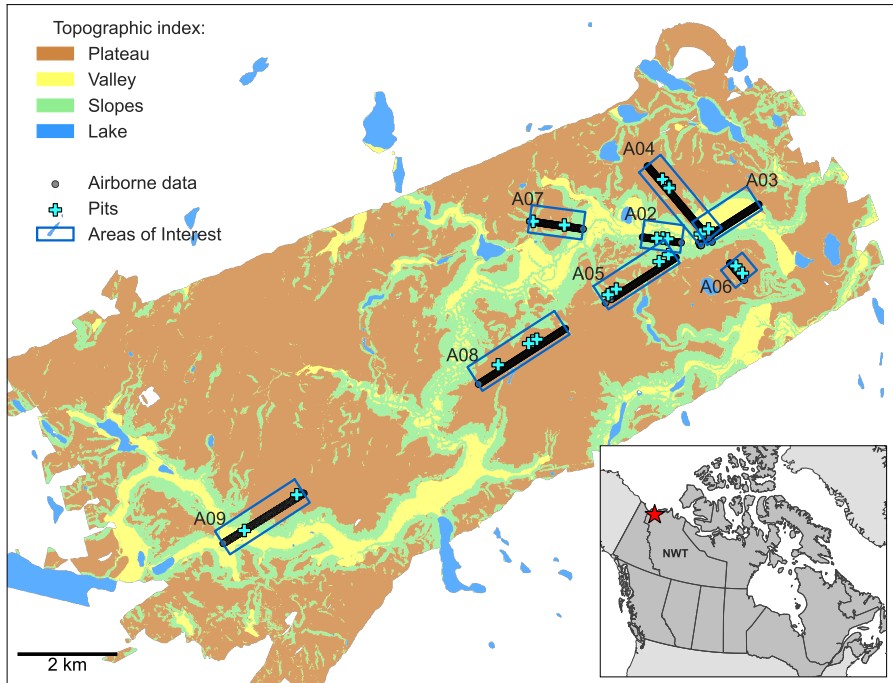

**Figure 1.** Topographic classification of Trail Valley Creek, NWT, Canada with locations of snow pits, Areas Of Interest (AOI), and an example of flight tracks within the AOIs. Inset demonstrates the location of TVC in the Northwest Territories, Canada. Adapted from Sandells et al. (2023).

## 2 Data and methods

### 2.1 MACSSIMIZE campaign

The MACSSIMIZE field campaign took place in Trail Valley Creek (TVC), NWT, Canada in March 2018, as part of the Year of Polar Prediction, an international, cross-disciplinary program of polar science studies coordinated by the Polar Prediction Project of the World Meteorological Organization (WMO). TVC consists of a mix of wind blown open tundra areas, and valleys with snow drifts and shrub vegetation, typical of Arctic tundra regions. Stratigraphy at the bottom of tundra snowpacks tends to be controlled by shrubs and underlying topography and strong temperature gradients, while layers above this are influenced by wind driven processes (Benson and Sturm, 1993; Sturm and Benson, 2004; Rutter et al., 2019).

Measurements focused on eight areas of interest (AOIs) as described in Sandells et al. (2023), the locations of which are shown in Fig. 1. The AOIs were chosen to capture the range of topographies, aspect and vegetation characteristics of TVC. Figure 1 also shows a map of broad topographic classifications for TVC; flat upland plateau, flat valley bottom and slopes (Rutter et al., 2019). Between 14-22 March, in situ snowpack measurements on the ground were co-located with flights of the Facility for Airborne Atmospheric Measurements (FAAM) BAe-146 research aircraft. This paper uses data from three flights





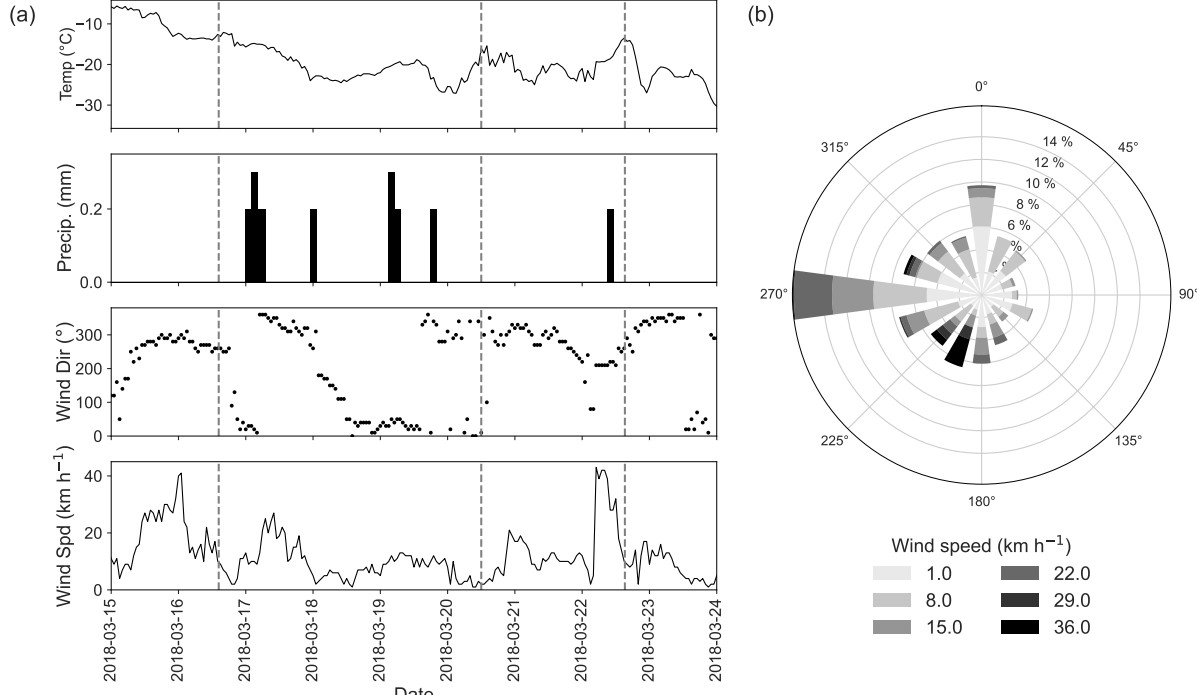

**Figure 2.** (a) Hourly meteorological data from Environment Canada Trail Valley station (WMO ID 71683) during the MACSSIMIZE campaign. Dashed lines indicate flight times at the start of TVC overpasses for C087 on 16 March, C090 on 20 March, and C092 on 22 March 2018. (b) Wind rose showing prevailing wind direction based on data in the third panel of (a).

(C087, C090 and C092) flown on 16, 20, and 22 March. Locations of snow pits where ground-based measurements were made, along with example flight tracks within the AOIs, are also shown in Fig. 1.

Precipitation and wind during the campaign caused changes in surface snow conditions between flights. Snowfall between
105 flights C087 and C090 introduced a low density fresh snow layer, which was evident in pits dug around this time. Increased wind speed before the third flight then redistributed the surface snow, altering snowpack stratigraphy. Freezing rain earlier in the season also introduced an ice layer to most snowpacks. Figure 2 shows how air temperature, precipitation, wind speed and wind direction changed during the campaign, as well as a wind rose demonstrating the prevailing wind direction. Vertical lines show the time of each flight.

**2.2 Airborne data and emissivity retrievals**

Airborne $T_b$ measurements were made using the Microwave Airborne Radiometer Scanning System (MARSS; McGrath and Hewison, 2001) and the International Submillimetre Airborne Radiometer (ISMAR; Fox et al., 2017) on board the FAAM aircraft. Both instruments are along track scanning radiometers containing dual-sideband heterodyne receivers measuring between 89 and 664 GHz. The position of these instruments on the side of the aircraft allows both upward and downward observations to



be made. On flights discussed in this paper, instruments were configured to make only downward-viewing observations during
AOI overpasses to increase the number of observations made over ground sites. Zenith and calibration views were obtained in
regions outside AOIs.

$T_b$ measurements at 89, 157 and 183 GHz from MARSS and 118 and 243 GHz from ISMAR allowed retrieval of emissivity
spectra. Frequencies above 243 GHz are not considered because the high atmospheric opacity means that satellite observations
at higher frequencies have little surface sensitivity, and emissivity retrievals can have large errors. The emissivity retrieval is
based on methods described in Harlow (2009), where observations of upwelling and downwelling brightness temperature close
to the surface are used to determine both emissivity and effective surface temperature using the relationship

$$T_{b,up,surf} = \epsilon T_{s,eff} + (1 - \epsilon)T_{b,down,surf}, \tag{1}$$

where $T_{b,up,surf}$ and $T_{b,down,surf}$ are surface upwelling and downwelling brightness temperatures, $T_{s,eff}$ is effective surface
temperature and $\epsilon$ is emissivity. Surface upwelling and downwelling brightness temperatures are determined from aircraft-level
observations by correcting for absorption and emission from the atmospheric layer between the aircraft and the surface. An
effective Lambertian value of $T_{b,down,surf}$ was derived from airborne observations at multiple zenith angles (Harlow, 2009),
as a Lambertian surface assumption has been shown to better represent snow surfaces at microwave frequencies over a specular
assumption (Guedj et al., 2010; Harlow and Essery, 2012; Bormann, 2022). Observations from multiple channels centred on a
130 gaseous absorption line (e.g. water vapour line at 183 GHz or oxygen line at 118 GHz) share the same emissivity and effective
temperature but will have different values of $T_{b,up,surf}$ and $T_{b,down,surf}$. This means both $\epsilon$ and $T_{s,eff}$ can be determined.
By assuming that $T_{s,eff}$ is approximately independent of frequency, Eq. (1) is then used to calculate $\epsilon$ at window channel
frequencies such as 89, 157 and 243 GHz.

MARSS and ISMAR scan patterns are not synchronised so observations from the two radiometers do not correspond to
135 exactly the same ground locations. Retrieved emissivities from the two instruments were mapped onto locations of ISMAR
observations by spatially averaging all observations where the beam centre location was within 50 metres of each ISMAR data
point. This resulted in 1916 observed emissivity spectra. An evaluation of the sensitivity of emissivity retrievals to errors in both
observed brightness temperatures and assumed parameters, such as temperature and humidity of the atmospheric layer between
the aircraft and the surface, suggests that retrieval error is approximately 0.01 for frequencies between 89 and 157 GHz, and
140 0.02 for frequencies between 183 and 243 GHz. These values are used as observation error in the SCEM-UA retrievals (Sect.
2.5). The larger error at higher frequencies is due to increased sensitivity to the amount of water vapour in the atmospheric
layer below the aircraft.

Retrieved emissivities were grouped using k-means clustering to give eight emissivity spectra, which are used as observa-
tions in the SCEM-UA retrievals. The number of clusters was chosen to represent variability in the retrieved emissivity while
providing distinct spectra from a significant number of observations. Figure 3 shows the eight emissivity spectra (coloured
lines), which are referred to as observations throughout the rest of the paper, and all observed spectra used to generate them
(grey lines). The range of spectral shapes reflect the variability in snow properties observed during the three flights. Clusters 0
and 2 were the only spectra where emissivity increased between 89 and 243 GHz, although all clusters show a slight decrease





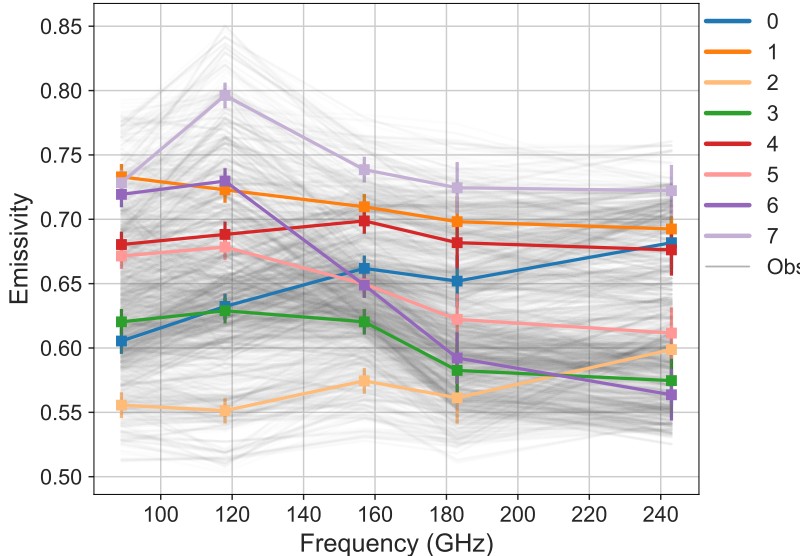

**Figure 3.** Emissivity spectra retrieved from airborne measurements (grey lines) and means of 8 groups (numbered 0-7) identified by k-means clustering.

in emissivity at 183 GHz. Emissivity values ranged from 0.55 to 0.80, with the lowest emissivity values associated with cluster

2 and the highest with cluster 7, which had a peak at 118 GHz. Tests were carried out to determine if this peak was caused by the Lambertian surface assumption in the emissivity retrieval, however, running the retrieval for a specular surface produced a similar spectrum. Given the steep change in emissivity it may be difficult for SMRT to simulate this spectra.

### 2.3   Snowpack measurements

Ground-based measurements of snow microstructure were made in 29 snow pits across eight AOIs. Snow pit locations were

aligned with the aircraft flight tracks, as shown in Fig. 1. Vertical profile measurements of density ($\rho_{snow}$ in kg m$^{-3}$), specific surface area (SSA in m$^2$ kg$^{-1}$) and temperature (K) were made at 3 cm vertical resolution in each of the snow pits, using techniques described in Rutter et al. (2019). In pits where two vertical profiles of density and SSA were measured, the average of two samples at each vertical position in the profile were used.

Individual layers within each snow pit were classified into three microstructure types, surface snow, wind slab and depth hoar,

based on their properties and through visual inspection (Rutter et al., 2019). Snow properties were averaged across the main microstructure types to quantify properties of each snowpack layer. All of the snow pits had a depth hoar layer, most contained wind slab, and several also had a fresh snow layer. Ice lenses were also present in most pits, and were included in simulations in Sandells et al. (2023), either by splitting layers, or adding the ice lens between layers depending where they occurred in the vertical profile. However, because this resulted in four or five layer snowpacks, this strategy would be too complex to introduce

into the MCMC methodology, which is already computationally intense, therefore ice lenses are not included in simulations





in this paper. Mean difference in simulated $T_b$ of snow pits from MACSSIMIZE with and without ice lenses was 1.88 K at 89 GHz, decreasing with frequency to 0.024 K at 243 GHz, so the impact of excluding ice lenses on emissivity simulations is expected to be minimal. The snow pit SSA, thickness, density and temperature observations for surface snow, wind slab and depth hoar are summarised in Table 1, and were used as input to SMRT as described in the following sections.

|  |  | Mean value | Min value | Max value |
|---|---|---|---|---|
| Correlation length (mm) | SS | 0.065 | 0.036 | 0.1 |
| | WS | 0.092 | 0.048 | 0.18 |
| | DH | 0.32 | 0.13 | 0.48 |
| Thickness (m) | SS | 0.062 | 0.03 | 0.12 |
| | WS | 0.12 | 0.0015 | 0.3 |
| | DH | 0.21 | 0.09 | 0.42 |
| Density (kg m⁻³) | SS | 94 | 38 | 250 |
| | WS | 310 | 200 | 420 |
| | DH | 260 | 200 | 350 |

**Table 1.** Snow parameter means and ranges used in SCEM-UA.

## 2.4 SMRT

SMRT was used to generate emissivity simulations, using sets of snow parameters generated during the MCMC retrieval as input (see Sect. 2.6.2), for two- and three-layer snowpacks, based on the three main observed snow layers. Three layers captured the main variations in snow profiles and observed stratigraphy, and matched layers simulated in Sandells et al. (2023). The underlying soil surface is assumed to be flat, with a temperature of 258.15K, as used in Sandells et al. (2023), and based on King et al. (2018).

SMRT requires layer thickness, density, temperature and a measure of grain size or microstructure depending on the microstructure model chosen. For this paper, SMRT was configured to use the Discrete Ordinate and Eigenvalue Solver (DORT) radiative transfer model, the Improved Born Approximation (IBA) electromagnetic model, and the exponential autocorrelation microstructure model. Although IBA is typically limited to lower frequencies than those in this paper, Picard et al. (2022) suggested the applicability of IBA could be extended to higher frequencies, and this was supported by good agreement between SMRT simulations and observations of $T_b$ at higher frequencies in Sandells et al. (2023). The exponential correlation length ($l_{ex}$) was used as the microstructure parameter, and was derived using observed SSA and density, according to the modified Debye relationship (Debye et al., 1957; Mätzler, 2002):

$$l_{ex} = \alpha_{db} \frac{4(1 - \rho_{snow}/\rho_i)}{SSA\rho_i} \qquad (2)$$



where $\rho_{snow}$ is snow density, $\rho_i$ is density of pure ice (916.7 kg m$^{-3}$), and $\alpha_{db}$ is the Debye modification parameter. A value

of 0.75 is used for $\alpha_{db}$ for surface snow and wind slab; a value proposed by Mätzler (2002). For depth hoar an $\alpha_{db}$ value of 1.2

is used, as proposed by Leinss et al. (2020), and discussed in Sandells et al. (2023).

SMRT does not directly simulate emissivity, therefore emissivity was retrieved by simulating upwelling brightness temper-

ature at the surface ($T_{bsmrt}$), for a viewing angle of 5 degrees, with different values for atmospheric downwelling ($T_{bdown}$)

according to Eq. (3). This is preferable to simply dividing $T_{bsmrt}$ by the physical snowpack temperature as snowpacks are not

isothermal and microwave penetration depths vary. This method is also used to derive emissivity in MEMLS (Wiesmann and

Mätzler, 1999).

$$\epsilon = 1 - \frac{T_b(T_{bdown} = 100K) - T_b(T_{bdown} = 0K)}{100K} \tag{3}$$

## 2.5 SCEM-UA overview

SCEM-UA is a global optimization routine, used to generate probability distributions of parameters of a model through ran-

dom sampling, using MCMC methods. SCEM-UA evolves multiple parallel Markov chains, using the Sequence Evolution

Metropolis (SEM) algorithm, which decides at each step in the Markov chain how likely a parameter value is to contribute

to the posterior probability distribution. SCEM-UA uses complex shuffling to share information about the parameter space

between chains to improve the efficiency and adaptability of the search compared to other MCMC samplers (Vrugt et al.,

2003).

SCEM-UA requires an estimation of the physical limits of parameters of interest (range of observed snow parameters; Table

1), a set of observed values to match (observed emissivity spectra; Fig. 3), an estimate of measurement error (0.01 from 89

to 157 GHz, and 0.02 from 183 to 243 GHz; Sect. 2.2) and a model for which the parameters are being estimated (SMRT).

SCEM-UA searches within parameter limits for parameter sets that give model simulations that match observed emissivity

values within measurement uncertainty. During this search SCEM-UA maps the probability distributions of the parameters

being searched. The returned posterior parameter distributions give an indication of parameter uncertainty associated with

simulations, and indicate the sensitivity of simulations to different snow parameters.

## 2.6 SCEM-UA–SMRT retrievals

SCEM-UA was coupled to SMRT and three experiments were run to assess the sensitivity of simulations to the different snow

parameters. The experiments were also split for two- and three-layer snowpacks to reflect the different stratigraphy observed

during MACSSIMIZE. Table 2 provides an overview of the three experiments, including the parameters being searched in each

snowpack layer (top panel), and the SCEM-UA setup for each experiment (bottom panel). More details on how the experiments

were split is given in Sect. 2.6.1 and 2.6.2.

In each experiment SCEM-UA was set up to run 10 parallel chains, with 50 initial samples per chain. Initial samples are

random values from a distribution between the minimum and maximum of the observed parameter range. The number of



| Experiment | Correlation length | | Correlation length & thickness | | Correlation length, thickness and density | |
|---|---|---|---|---|---|---|
| Clusters | 0, 1, 2, 4, 7 | 3, 5, 6 | 0, 1, 2, 4, 7 | 3, 5, 6 | 0, 1, 2, 4, 7 | 3, 5, 6 |
| SS | - | CL | - | CL TH | - | CL TH D |
| WS | CL | CL | CL TH | CL TH | CL TH D | CL TH D |
| DH | CL | CL | CL TH | TH | CL D | |
| | | | | | | |
| Chains | 10 | | 10 | | 10 | |
| Initial samples | 50 | | 50 | | 50 | |
| Iterations | 10,000 | | 20,000 | | 20,000 | |
| Burn-in | 2000 | | 5000 | | 12,000 | |

**Table 2.** Setup of three SCEM-UA-SMRT experiments. Top panel shows parameters varied in each layer for the two- and three-layer clusters. CL is correlation length, TH is thickness, and D is density. Dashes denote that surface snow was not present in two-layer snowpacks (see Sect. 2.6.1). Empty cell shows that depth hoar parameters were fixed for three-layer clusters in the third experiment (see Sect. 2.6.2). Bottom panel shows SCEM-UA parameter setup for each experiment. The number of initial samples and iterations are per chain. Burn-in represents the number of samples removed before chains converged based on the $\hat{R}$ diagnostic, as described in Sect. 2.6.

iterations per chain varied between 10,000 and 20,000 depending on the number of parameters and snowpack layers allowed to vary in the retrieval, and how many iterations were needed for the chains to converge. Convergence was assessed using a combination of the rank normalised $\hat{R}$ convergence criteria (Vehtari et al., 2021) and auto-correlation. The rank normalised $\hat{R}$ convergence diagnostic compares variance between chains $\hat{V}$ to variance within each chain $W$ and is computed by:

$$\hat{R} = \frac{\hat{V}}{W} \qquad (4)$$

When $\hat{R}$ reaches 1.0, the between-chain and within-chain variance are equal and chains have successfully converged, however a value of less than 1.01 is deemed sufficient to indicate convergence. Samples taken before $\hat{R}$ is below 1.01 are removed from the beginning of the evolution, and the number of iterations this takes are given by the burn-in number in Table 2. Samples taken after chains have converged represent samples from the posterior distribution of parameter sets, which are presented in

Sect. 3.2. Each parameter set within the posterior distribution is then used to calculate a spectrum using SMRT. The means and standard deviations of these spectra are presented in Sect. 3.1. Comparisons of simulated and observed emissivity spectra allowed an assessment of SMRT's ability to reproduce the emissivity spectra within observed parameter limits, while the sensitivity of simulations to each parameter was assessed based on the posterior parameter distributions.

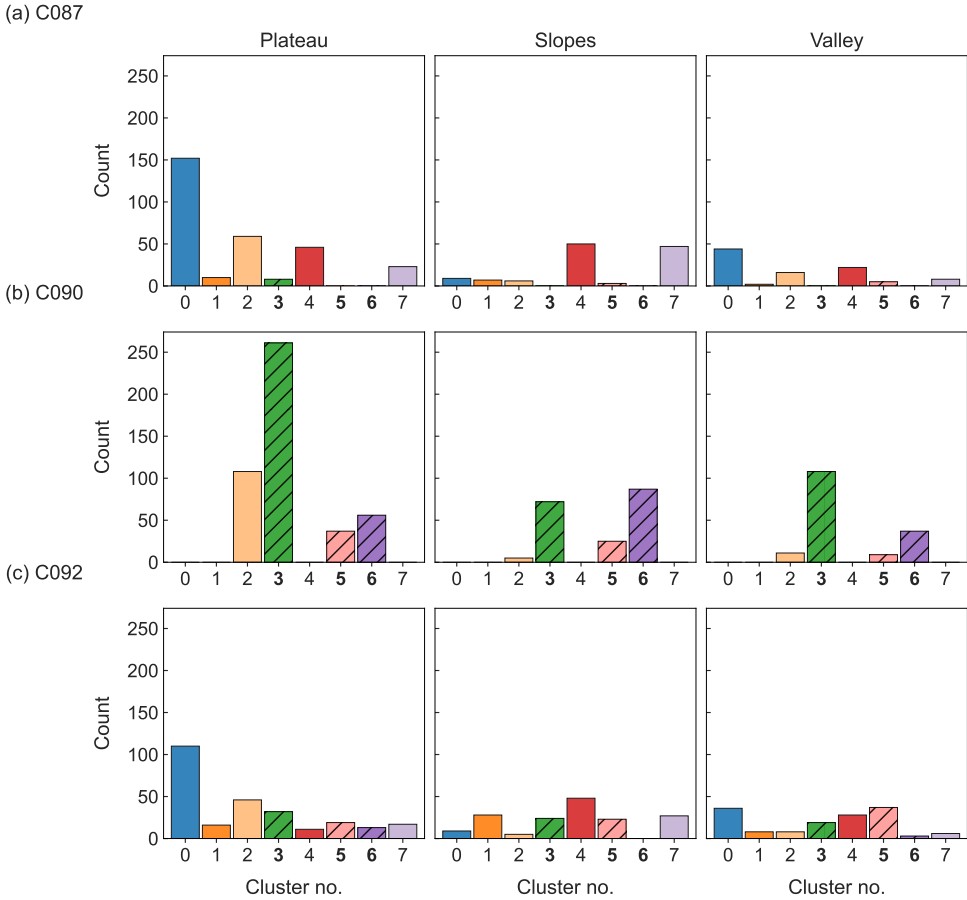

**Figure 4.** Count of emissivity clusters observed over the three flights and topographic groups of TVC. Colours match spectra introduced in Fig. 3. Solid bars show clusters simulated as two-layer snowpacks (0, 1, 2, 4 and 7); hatched bars and bold labels show clusters simulated as three-layer snowpacks (3, 5 and 6).

### 2.6.1 Splitting retrievals by snowpack layers

Observed emissivity varied during the campaign due to changes in meteorological conditions, and subsequent changes in snowpack stratigraphy and microstructure. Figure 4 shows the distribution of observed emissivity clusters between the three flights and three main topographic classifications. Clusters 0, 2, 4 and 7 were observed most often in flight C087, with 0 observed mostly in plateau and valley regions, and 4 and 7 dominating sloped regions. In flight C090 the predominant clusters were 3 and 6, with 3 observed mostly in plateau regions, and 6 in sloped regions. None of the clusters from C087 were observed

in C090 except cluster 2. The difference in observed emissivity between C087 and C090 was due to snowfall between the flights, which introduced a low density surface snow layer. Increased winds between C090 and C092 redistributed the surface snow, resulting in the greater variability in spectra observed in flight C092.





To reflect these differences the retrievals were split into two groups. Retrievals for clusters 0, 1, 2, 4 and 7, which were observed more often in C087, were run using a two-layer snowpack (wind slab and depth hoar). Clusters 3, 5 and 6, which were observed more often during C090, were run using a three-layer snowpack (surface snow, wind slab and depth hoar), to reflect the addition of the surface snow layer. Table 2 shows how the experiments were split for the two- and three-layer clusters.

### 2.6.2 Splitting retrievals by snow parameters

Correlation length, layer thickness and density were chosen as parameters that were expected to significantly impact emissivity simulations. Correlation length represents grain size and influences scattering, along with density, while layer thickness and scattering strength influence the penetration depth of microwaves. Changing penetration depth with frequency results in variable spectral shapes due to variations in microstructure with depth. Total snowpack depth also influences sensitivity of microwave sensors to substrate below. In the first experiment, only correlation length was varied, then correlation length and layer thickness in the second, and correlation length, layer thickness and density in the third. In the second experiment, depth hoar correlation length was fixed for the three-layer clusters to improve efficiency. This was chosen due to low sensitivity to this parameter identified in the first experiment. In the third experiment depth hoar thickness of the two-layer clusters, and all depth hoar parameters for the three-layer clusters were fixed. The basis for this free parameter reduction is discussed more in Sect. 3.2. Table 2 outlines parameters that were varied in each snowpack layer for the three experiments.

The parameter limits which SCEM-UA searches within during the retrieval were provided by observed snow parameter ranges across all the snow pits (Table 1). Parameters not included in the retrievals, as well as snowpack temperature, were fixed at the mean value from all the pits. During the second experiment, the original upper limit of the wind slab thickness range was found to be too large at 0.71 m, resulting in chains failing to converge and returned parameter values falling at the lower end of the limits. The range was therefore restricted by removing outliers from pits dug in deeper snow drifts that were end members of the snow depth distribution. This gave an upper limit for wind slab thickness of 0.3 m.

## 3 Results

### 3.1 SMRT vs observations

The mean SMRT simulated emissivity spectra for parameter sets retrieved after chains had converged are shown in Fig. 5 (standard deviation shown by error bars) for the three experiments, along with observed emissivity spectra for each cluster. Table 3 gives the root mean square error (RMSE) and mean absolute error (MAE) between observed and simulated spectra for each cluster, in each experiment.

When only correlation length was varied by SCEM-UA in the first experiment, only clusters 1 and 5 had mean simulated emissivities which fell within observation error at all frequencies (green lines in Fig. 5). For clusters 3, 4, 6 and 7, SMRT reproduced the shape of the spectra reasonably well, for example, both observed and simulated emissivity decreased overall





**Figure 5.** SMRT simulated mean and standard deviation of emissivity spectra from the three experiments, compared with observed spectra from Fig. 3 for each cluster. Shading denotes observation error used in SCEM-UA (Sect. 2.2). CL is correlation length, TH is thickness, and D is density.

between 118 and 243 GHz for clusters 3, 6 and 7, and increased between 89 and 118 GHz for clusters 4 and 7. However, mean simulated emissivities fell outside observation error at some frequencies (89 GHz for cluster 3, 157 GHz for cluster 4, 183 GHz for cluster 6, and 118 GHz for cluster 7). For cluster 7, this relates to the peak in the observed spectrum mentioned in Sect. 2.2. SMRT was not able to simulate observed spectra of clusters 0 and 2 when only correlation length was allowed to vary. The slope of the simulated spectrum of cluster 0 was opposite to that of the observed spectrum up to 183 GHz, and although the shape of the cluster 2 spectrum was similar to observations, emissivity values were too high, with an MAE of 0.063.



| | Correlation length | | Correlation length & thickness | | Correlation length, thickness & density | |
|---|---|---|---|---|---|---|
| | RMSE | MAE | RMSE | MAE | RMSE | MAE |
| 0 | 0.028 | 0.025 | **0.011** | 0.0061 | 0.012 | **0.0052** |
| 1 | **0.0081** | **0.0034** | 0.0086 | 0.0041 | 0.011 | 0.0037 |
| 2 | 0.064 | 0.063 | 0.017 | 0.011 | **0.013** | **0.0069** |
| 3 | 0.018 | 0.015 | **0.013** | 0.0084 | 0.014 | **0.0056** |
| 4 | 0.012 | 0.0093 | **0.01** | **0.0055** | 0.012 | 0.0059 |
| 5 | 0.013 | 0.0079 | 0.013 | 0.0077 | **0.012** | **0.0057** |
| 6 | 0.016 | 0.011 | **0.013** | **0.0074** | 0.014 | 0.0084 |
| 7 | 0.017 | 0.013 | **0.016** | **0.012** | 0.017 | 0.013 |
| Overall | 0.029 | 0.018 | **0.013** | 0.0078 | **0.013** | **0.0068** |

**Table 3.** RMSE and MAE between observed and simulated emissivity spectra for each cluster, in the three experiments, for simulations produced using parameters after chains had converged. Bold text indicates minimum RMSE and MAE for each cluster.

RMSE (and MAE) averaged across all clusters and frequencies, decreased from 0.029 (0.018) to 0.013 (0.0078) when layer thickness was also accounted for. Simulated emissivities were within observation error at all frequencies for clusters 0, 1, 3, 4, 5 and 6, compared to only clusters 1 and 5 in the first experiment. The biggest improvement was for clusters 0 and 2, where SMRT was not able to reproduce the observed spectra with correlation length alone. Only clusters 2 and 7 had mean simulated emissivities which fell outside observation errors, at 183 and 243 GHz for cluster 2, and 118 GHz for cluster 7. Clusters 2 and 7 also had the largest RMSE (and MAE) in the second experiment, at 0.017 (0.011) and 0.16 (0.012) respectively. The only case where the standard deviation of the simulation did not overlap with the observation error was cluster 7 at 118 GHz. Although mean simulated emissivity at 118 GHz increased slightly when layer thickness (and density in the third experiment) were accounted for, this was still not enough to reproduce the peak in the observed spectrum.

Overall RMSE and MAE did not change much between the second and third experiment, when density was also varied. Most clusters had simulated emissivities within observation error at all frequencies, except cluster 6, where simulated emissivity at 183 GHz was just outside the upper observation error, and cluster 7. Most changes in simulated emissivity in the third experiment were seen at 243 GHz, indicating sensitivity to density in the surface snow, due to the shallower penetration depth at this frequency. For clusters 0, 2, 3 and 5, simulated emissivities at 243 GHz were a better match to observations in the third experiment, most notably for cluster 2 where simulations at 183 and 243 GHz previously fell outside observation error (although the shape of the spectra was a worse match for observations at lower frequencies). For clusters 1, 4 and 7, there was slightly less agreement between simulations and observations. However, these changes in emissivity were relatively small, and the standard deviation of simulations was larger at 243 GHz than in previous experiments, shown by larger error bars in Fig. 5. This could be due to 243 GHz being outside the limit of applicability of the IBA electromagnetic model (see Sect. 2.4).



Overall, Fig. 5 suggests that SMRT is capable of simulating a variety of emissivity spectra between 89 and 243 GHz, and demonstrates the importance of representing at least the correlation length and thickness of different snow layers. In the last two experiments, the MAE for most clusters were smaller than the lower observation error of 0.01 (Sect. 2.5).

## 3.2 Parameter posterior distributions

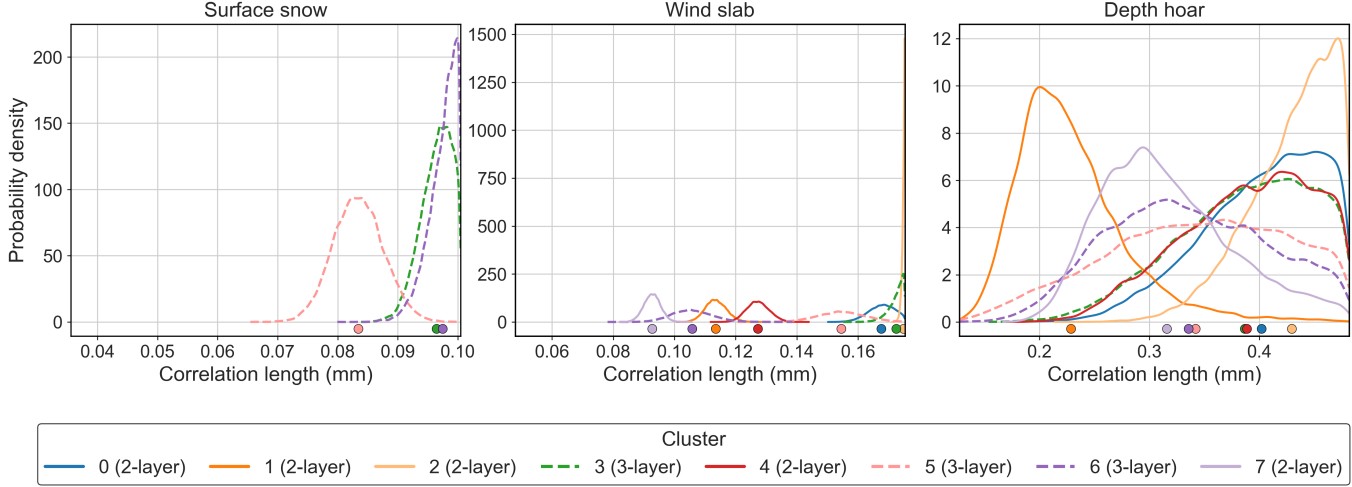

**Figure 6.** Posterior parameter distributions for correlation length (mm) of surface snow, wind slab and depth hoar in the first experiment. Two-layer clusters are shown by solid lines, three-layer clusters are shown by dashed lines. X-axis limits represent parameter ranges from snow pit observations used to constrain the retrievals. Dots indicate the mean of each distribution.

Probability distributions for parameter sets retrieved by SCEM-UA after chains had converged in each experiment are shown in Figs. 6-8. Table 4 gives the retrieved mean and standard deviation for these distributions. As described in Sect. 2.6.1, simulations for clusters 0, 1, 2, 4 and 7 were run as two-layer snowpacks, and clusters 3, 5 and 6 as three-layer snowpacks, hence surface snow parameters are only given for 3, 5 and 6.

Figure 6 shows distributions for correlation length in the first experiment. Several clusters had narrow distributions with low standard deviation for surface snow (cluster 5) and wind slab (clusters 1, 4, 6 and 7), indicating sensitivity of simulations to correlation length in these layers. In comparison, depth hoar distributions for most clusters were relatively broad and spanned most of the parameter range, with higher standard deviation than the other layers (0.039 mm to 0.081 mm; Table 4), suggesting simulations were less sensitive to depth hoar. Low sensitivity to depth hoar is expected given relatively shallow penetration depths at higher frequencies, and the need for depth hoar emissions to penetrate the layers above. Given this low sensitivity, and the increased number of parameters being searched in subsequent experiments, depth hoar correlation length was fixed at the mean for the three layer clusters in the second and third experiments (Sect. 2.6.2).

Several distributions in all three layers were at the upper limits of parameter ranges, such as surface snow correlation length of clusters 3 and 6, wind slab correlation length of clusters 0 and 2 (where SMRT was not able to reproduce the observed spectra),





**Figure 7.** Posterior parameter distributions for correlation length (mm) and thickness (m) of surface snow, wind slab and depth hoar in the second experiment. Two-layer clusters are shown by solid lines, three-layer clusters are shown by dashed lines. X-axis limits represent parameter ranges from snow pit observations used to constrain the retrievals. Dots indicate the mean of each distribution.

and depth hoar correlation length of clusters 0, 2, 3 and 4. This, along with the inability of SMRT to reproduce observed emissivities at all frequencies (Sect. 3.1), suggests that either correlation length ranges from observed snow parameters were not large enough, or that varying correlation length alone is not sufficient to allow SMRT to simulate observations when other

parameters are fixed.

Figure 7 shows distributions from the correlation length and thickness experiment. Correlation length distributions for surface snow and wind slab did not change much when thickness was also varied. The exceptions were wind slab correlation length distributions of clusters 0, 2 and 6. For cluster 2, wind slab correlation length was at the upper limit in the first experiment, but was spread across the parameter range in the second, suggesting low sensitivity to the wind slab layer for this cluster. This can

be explained by wind slab thickness for cluster 2 falling at the bottom of the parameter range (mean thickness of 0.0047 m), indicating a very shallow or non-existent layer. Shallow wind slab resulted in increased sensitivity to depth hoar, indicated by



a narrow distribution for depth hoar correlation length with standard deviation of 0.014 mm, much lower than that of the other depth hoar distributions. Greater sensitivity to large grained depth hoar (larger correlation lengths) caused increased scattering, and contributed to the low emissivities of the cluster 2 spectrum, which could not be simulated without representing wind slab thickness.

With the exception of cluster 2, relatively narrow wind slab thickness distributions of clusters 0, 3, 4, 6 and 7, indicate sensitivity of simulations to wind slab thickness, as well as correlation length. Surface snow thickness distributions had similar standard deviations to wind slab (0.016 m to 0.024 m), however, because the observed surface snow thickness range was relatively small (0.03 m to 0.12 m, Table 1), distributions spanned the parameter range. Sensitivity to the thickness and correlation length of surface snow and wind slab is expected, as the thickness and scattering strength of layers near the top of the snowpack influence attenuation of emission from lower layers, as seen for cluster 2. Unlike cluster 2, parameter distributions suggested that cluster 7 was associated with a relatively deep wind slab layer (0.14 m) with relatively low correlation length (0.094 mm). Lower correlation length relates to smaller snow grains, resulting in reduced scattering across a relatively deep layer, which reduced penetration to depth hoar, and resulted in higher emissivities across the cluster 7 spectrum.

Depth hoar thickness distributions spanned the full parameter range for all clusters, including cluster 2, and had higher standard deviation than the other layers (0.083 m to 0.09 m), again indicating low sensitivity of simulations to depth hoar thickness. Even for shallow thicknesses, the higher frequencies are unlikely to penetrate through the depth hoar to the bottom of the snowpack, meaning changes in thickness of this layer are less important. Therefore, in the final experiment depth hoar thickness was fixed at the mean, in addition to the correlation length, to reduce the number of parameters being searched.

Figure 8 shows distributions for the correlation length, thickness and density experiment. Wind slab parameter distributions were broader with higher standard deviations (Table 4) than in previous experiments, for most clusters. This is likely due to the complex interaction of snowpack microstructure parameters such as correlation length and density, whereby a range of combinations of density and correlation length could result in similar scattering properties. Simulations were therefore similarly sensitive to both correlation length and density. Wind slab density distributions were also relatively broad, with standard deviation ranging from 22 to 58 kg m$^{-3}$, and as in the other experiments, depth hoar distributions spanned the parameter ranges.

Surface snow correlation length and density distributions had lower standard deviations than wind slab. For all three-layer clusters, surface snow correlation lengths were at the upper end of the parameter range, with mean retrieved values between 0.088 and 0.096 mm, similar to the lowest wind slab correlation lengths of clusters 6 and 7. These distributions did not change much between the three experiments. Surface snow density distributions were at the lower end of the range, with mean retrieved density between 86 and 120 kg m$^{-3}$ (Table 4). Clusters with surface snow at the top of the snowpack (3, 5 and 6) had lower emissivities at 183 and 243 GHz than those with wind slab at the top of the snowpack (except cluster 2). For example, clusters 6 (three-layer) and 7 (two-layer) had similar emissivity at 89 GHz (0.72 and 0.73) due to similar wind slab scattering properties, but at 243 GHz the emissivity of cluster 6 (0.56) was lower than cluster 7 (0.72). Low density, and relatively low correlation lengths in the surface snow layer should result in less scattering and higher, rather than lower, emissivity at the higher frequencies. Another explanation for the differences in emissivity caused by surface snow is boundary effects caused by dielectric contrasts between layers of different density, this is discussed further in Sect. 4.





**Figure 8.** Posterior parameter distributions for correlation length (mm), thickness (m) and density (kg m$^{-3}$) of surface snow, wind slab and depth hoar in the third experiment. Two-layer clusters are shown by solid lines, three-layer clusters are shown by dashed lines. X-axis limits represent parameter ranges from snow pit observations used to constrain the retrievals. Dots indicate the mean of each distribution. Depth hoar thickness was fixed for all clusters in the third experiment so no distributions are shown.



| Experiment | | Correlation length (mm) | | Correlation length (mm) | | Thickness (m) | | Correlation length (mm) | | Thickness (m) | | Density (kg m⁻³) | |
|---|---|---|---|---|---|---|---|---|---|---|---|---|---|
| Cluster | Snow layer | Mean | Std | Mean | Std | Mean | Std | Mean | Std | Mean | Std | Mean | Std |
| 0 | WS | 0.17 | 0.0042 | 0.13 | 0.011 | 0.045 | 0.011 | 0.13 | 0.024 | 0.039 | 0.013 | 310 | 39 |
|  | DH | 0.4 | 0.054 | 0.32 | 0.066 | 0.26 | 0.087 | 0.32 | 0.067 | - | - | 280 | 40 |
| 1 | WS | 0.11 | 0.0035 | 0.11 | 0.0041 | 0.15 | 0.039 | 0.13 | 0.017 | 0.16 | 0.051 | 340 | 41 |
|  | DH | 0.23 | 0.05 | 0.29 | 0.094 | 0.26 | 0.087 | 0.3 | 0.095 | - | - | 270 | 40 |
| 2 | WS | 0.17 | 0.00048 | 0.13 | 0.029 | 0.0047 | 0.0023 | 0.15 | 0.02 | 0.034 | 0.023 | 240 | 22 |
|  | DH | 0.43 | 0.039 | 0.24 | 0.014 | 0.27 | 0.083 | 0.3 | 0.066 | - | - | 260 | 33 |
| 3 | SS | 0.096 | 0.0026 | 0.096 | 0.0028 | 0.081 | 0.022 | 0.091 | 0.0063 | 0.077 | 0.023 | 86 | 26 |
|  | WS | 0.17 | 0.0025 | 0.15 | 0.022 | 0.046 | 0.015 | 0.15 | 0.016 | 0.093 | 0.047 | 250 | 33 |
|  | DH | 0.39 | 0.061 | - | - | 0.25 | 0.09 | - | - | - | - | - | - |
| 4 | WS | 0.13 | 0.0037 | 0.12 | 0.0059 | 0.083 | 0.021 | 0.13 | 0.021 | 0.082 | 0.026 | 310 | 45 |
|  | DH | 0.39 | 0.059 | 0.29 | 0.083 | 0.26 | 0.088 | 0.29 | 0.082 | - | - | 280 | 41 |
| 5 | SS | 0.083 | 0.0042 | 0.084 | 0.0047 | 0.072 | 0.024 | 0.088 | 0.0078 | 0.076 | 0.024 | 120 | 38 |
|  | WS | 0.15 | 0.0077 | 0.15 | 0.014 | 0.13 | 0.054 | 0.14 | 0.018 | 0.13 | 0.054 | 270 | 41 |
|  | DH | 0.34 | 0.081 | - | - | 0.25 | 0.087 | - | - | - | - | - | - |
| 6 | SS | 0.097 | 0.0024 | 0.096 | 0.0031 | 0.096 | 0.016 | 0.096 | 0.0033 | 0.096 | 0.016 | 97 | 23 |
|  | WS | 0.11 | 0.0066 | 0.091 | 0.016 | 0.11 | 0.019 | 0.094 | 0.02 | 0.13 | 0.04 | 300 | 58 |
|  | DH | 0.34 | 0.071 | - | - | 0.26 | 0.086 | - | - | - | - | - | - |
| 7 | WS | 0.093 | 0.0027 | 0.094 | 0.0028 | 0.14 | 0.012 | 0.088 | 0.0082 | 0.16 | 0.034 | 270 | 41 |
|  | DH | 0.32 | 0.059 | 0.38 | 0.066 | 0.26 | 0.084 | 0.38 | 0.064 | - | - | 260 | 39 |

**Table 4.** Mean and standard deviation (std) of probability distributions for correlation length (mm), thickness (m) and density (kg m⁻³) for surface snow (SS), wind slab (WS) and depth hoar (DH) where they were present for each cluster, in the three experiment. Dashes show where parameters were fixed in retrievals and therefore no distributions were returned.



## 4 Discussion and conclusions

The aims of this paper were to assess the potential for SMRT to model snow surface radiative transfer for NWP, and evaluate emissivity simulations between 89 and 243 GHz to identify which parameters must be represented by a surface model to achieve accurate emissivity simulations. SMRT was coupled to the SCEM-UA MCMC algorithm to retrieve snow parameters from a set of observed emissivity spectra, and to compare observed and SMRT simulated emissivities. Physically realistic snow parameter ranges for correlation length, layer thickness and density were derived from observations from 29 snow pits in Trail Valley Creek. The results of the SCEM-UA retrievals showed that SMRT is capable of simulating a range of observed emissivity spectra. Varying correlation length allowed SMRT to capture much of the variability in observed emissivity, however, SMRT was not able to reproduce the full range of emissivity spectra without also varying the thickness of different snow layers. RMSE decreased from 0.029 to 0.013, and MAE from 0.018 to 0.0078, when thickness was varied by SCEM-UA along with correlation length, compared to correlation length alone.

Correlation length is a measure of snowpack microstructure, which relates to grain size and therefore influences scattering. The importance of representing heterogeneous profiles of grain size for modelling microwave brightness temperature and emissivity has been shown in many studies (e.g. Armstrong et al., 1993; Grody, 2008; Brucker et al., 2010; Harlow and Essery, 2012; Rutter et al., 2014). Under a Rayleigh scattering assumption, scattering is determined by the ratio of wavelength to grain size, and increases with increasing grain size (and increasing density). Smaller correlation lengths correspond to smaller snow grains, less scattering and higher emissivity, for example, high emissivities of cluster 7 were attributed to a deep weakly scattering wind slab layer with low correlation length. Scattering also increases with frequency, although this assumption is limited at higher frequencies, particularly for coarse grained snow such as depth hoar, when the wavelength becomes comparable to grain size. This should be considered when modelling at 243 GHz, as even with extended frequency limits (see Sect. 2.4; Picard et al., 2022; Sandells et al., 2023), the IBA electromagnetic model may not be applicable at this frequency.

Simulations in the third experiment were similarly sensitive to correlation length and density, due to the influence of both parameters on scattering. The impact of density was most obvious in the surface snow layer. The three-layer clusters (3, 5 and 6) had lower emissivities at the higher frequencies, which are more sensitive at the top of the snowpack. Surface snow distributions indicated densities as low as 86 kg m$^{-3}$, which relates to the low density surface snow layer introduced by snowfall between flights C087 and C090. Most of the observed snow pits had densities below 100 kg m$^{-3}$. These values are similar to the fresh snow density used in the current JULES snow scheme of 109 kg m$^{-3}$ (Walters et al., 2019). Although the low density (and low correlation lengths) of the surface snow indicates reduced scattering, Sandells et al. (2023) suggested $T_b$ differences between flights resulting from this low density surface layer were instead caused by high density-driven dielectric contrast between layers. The difference in emissivity spectra of clusters with and without surface snow highlights the importance of capturing different snowpack stratigraphy, and the impact of both density and correlation length on the emissivity spectra suggests both parameters need to be represented for emissivity modelling. However, it is also important to know the thickness of these layers, due to the impact of thickness on penetration to deeper layers.



Distributions of parameter sets suggested that simulations were most sensitive to wind slab parameters, and surface snow where it was present, with less sensitivity to depth hoar. Depth hoar is still needed for simulating the emissivity of Arctic snowpacks, as depth hoar is a key feature of tundra snow (Sturm et al., 1995). However, at higher frequencies, as snowpack depth increases or the strength of scattering in higher layers increases, sensitivity to deeper layers of the snowpack decrease due to increased scattering and absorption (Sturm et al., 1995; Brucker et al., 2010; Saberi et al., 2017). In deeper snowpacks with substantial surface snow or wind slab, using average parameter values for depth hoar may be sufficient for modelling emissivity at these frequencies, however, it is important to know both the thickness and scattering strength of overlying layers, which control penetration to deeper snow. Cluster 2, for example, had the greatest sensitivity to depth hoar due to shallower overlying wind slab than the other clusters. The current JULES snow scheme models three snowpack layers, and snow accumulates to maximum thicknesses of 0.04 and 0.12 m in the top two layers of the snowpack, with subsequent snow accumulating in the bottom layer (Walters et al., 2019). Future work should consider the suitability of existing snowpack model configurations, such as that in JULES, for emissivity modelling.

The high spatial and temporal variability in emissivity observed during MACSSIMIZE, relating to changes in snowpack stratigraphy and microstructure, highlights the potential for using modelled emissivities in place of a fixed emissivity atlas. Snowpack depth and stratigraphy are influenced by meteorology, as seen in the impacts of snowfall and wind, and by the interaction of snow with topography and vegetation (Liston and Sturm, 1998; Winstral et al., 2002; Sturm and Benson, 2004). Therefore, in order to model emissivity, snow parameter estimation from a land surface model is needed. When considering the scale of satellite footprints, e.g. 16 km at nadir for MHS (EUMETSAT, 2023b) and 17 km for the 89 and 183 GHz channels on the Microwave Sounder (MWS) (EUMETSAT, 2023c), averaging parameter values across model grid boxes may be sufficient (e.g. Armstrong et al., 1993), as discussed in relation to depth hoar at higher frequencies. However, other methods could be utilised that represent snow heterogeneity using a variability parameter, which may be based on changes in topography, vegetation, or land cover fractions (e.g. Liston, 2004; Sturm and Wagner, 2010; Derksen et al., 2012; Rutter et al., 2019; Meloche et al., 2022), by taking the mean and variance from distributions of parameters, such as those produced in this paper.

This paper demonstrates the potential for SMRT to simulate a variety of emissivity spectra between 89 and 243 GHz when supplied with snowpack stratigraphy information, such as the thickness and scattering properties of snowpack layers. Correlation length, layer thickness and density were all assessed in the retrievals, and it was shown that, as a minimum, variability in both correlation length and layer thickness, particularly for wind slab, needed to be captured in order to match all the observed emissivity spectra. This is an important step to improve the assimilation of satellite microwave data in Arctic regions for NWP. Future work will aim to assess the impact of SMRT emissivities on the simulation of top of atmosphere brightness temperatures in an integrated forecasting system. This system will require a physical surface snow model that is able to simulate snowpack properties required by SMRT, including grain size, density and layer thickness, particularly in the top layers of the snowpack.

*Data availability.* Retrieved emissivity spectra are available from Zenodo (Wivell et al., 2023). Airborne data for the MACSSIMIZE campaign are available from the Centre for Environmental Data Analysis (Facility for Airborne Atmospheric Measurements et al., 2018). Me-



teorological data can be downloaded from https://climate.weather.gc.ca/historical_data/search_historic_data_e.html for Trail Valley station
(WMO ID 71683), March 2018.

*Author contributions.* KW designed and performed the analysis with input from SF. MS contributed to design of SMRT simulations. CH, NR and RE planned and coordinated the combined ground and airborne campaign. SF and CH made airborne observations. NR and RE made ground-based observations. KW prepared the manuscript with contributions from all co-authors.

*Competing interests.* The authors declare that there are no competing interests.

*Acknowledgements.* Data collection was made possible thanks to NERC Arctic Office UK and the Canada Arctic Partnership Bursaries Programme (to NR and RE), Wilfrid Laurier University (Phil Marsh and Branden Walker) and Environment and Climate Change Canada. Thanks to Peter Toose (Environment and Climate Change Canada) for support creating the topographic classification of Trail Valley Creek. Radiometric surface-based measurements were supported by the Natural Sciences and Engineering Research Council of Canada (NSERC) and by Polar Knowledge Canada. Thanks to Alain Royer (Université de Sherbrooke) and Alexandre Roy (Université du Québec à Trois-
Rivières) for ground snow data collection. Airborne data was obtained using the BAe-146-301 Atmospheric Research Aircraft [ARA] flown by Airtask Ltd and managed by FAAM Airborne Laboratory, jointly operated by UKRI and the University of Leeds. Thanks to members of Observation-based Research at the Met Office for supporting airborne radiometer data collection. The MACSSIMIZE campaign was part of the Year of Polar Prediction effort, coordinated by the WMO Polar Prediction Project.



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
