# Peer review of "Evaluating Snow Microwave Radiative Transfer (SMRT) model emissivities with 89 to 243 GHz observations of Arctic tundra snow"

_EGUsphere, 2023_

## Author Response (AR1)

Dear Editor,

Please find below the point-by-point responses to referee comments, including the changes made in the revised manuscript. Referee comments are given in black text, responses and changes are given in blue text.

Line numbers in the referee comments relate to the original manuscript, however, when highlighting changes made, line numbers are relevant to the revised version of the manuscript.

Regards,

On behalf of the co-authors,

Kirsty Wivell
* * *
**Response to Referee #1**

General

This is an interesting paper that uses passive microwave observations from aircraft to assess to what extent SMRT is able to model emissivity spectra using snow parameters that are within observed values from snow pits. The paper is very well written, with methods and results described concisely and thoroughly. As the paper states, the results are relevant in the context of better assimilation of passive microwave observations from satellites over snow-covered areas. The paper can be accepted subject to addressing the minor comments below.

Minor comments

**Comment 1:**

Section 2.1: How do the snowpack measurements relate to the aircraft measurements in terms of timing? It is mentioned that the snowpack measurements were taken between 14-22 March 2018, with the flights taking place on 16, 20, 22 March 2018. Since the paper mentions considerable changes in the snow pack between flights, the temporal collocation appears relevant.

**Response 1:**

The dates on which pits were dug are given in the table below. The pits were dug on several days between the 14-22 March, with some dug on the same day as a flight, and others on days between flights. We have quantified the expected impact of snow pit timings in Sandells et al. (2023, preprint). The benefit of the retrieval method here, using the range of parameters observed across all the snow pits, and clustering the observations from all three flights together, is that it captures any changes in the airborne and ground-based observations resulting from changing conditions between flights. Designing the retrievals for two- and three-layer snowpacks allowed us to represent the key difference in the snowpacks, as discussed in Sect. 2.6.1.

| A02W | 14/03/2018 | A09E | 20/03/2018 |
|------|------------|------|------------|
| A02C | 15/03/2018 | A08W1 | 20/03/2018 |
| A02E | 15/03/2018 | A08W | 20/03/2018 |
| A04S1 | 16/03/2018 | A08E | 20/03/2018 |
| A04S | 16/03/2018 | A08C | 20/03/2018 |
| A04N | 16/03/2018 | A05E | 20/03/2018 |
| A04C1 | 16/03/2018 | A05W | 20/03/2018 |
| A04N1 | 16/03/2018 | A09W | 20/03/2018 |
| A04C | 16/03/2018 | A05C1 | 20/03/2018 |
| A03W | 17/03/2018 | A07W | 21/03/2018 |
| A03E | 17/03/2018 | A05N | 21/03/2018 |
| A03C1 | 17/03/2018 | A05C | 21/03/2018 |
| A06N | 18/03/2018 | A07C | 21/03/2018 |
| A06C | 18/03/2018 | MetS | 22/03/2018 |
| A06S1 | 18/03/2018 | | |

**Comment 2:**

L 132-133: The assumption is made that T_s,eff is approximately independent of frequency. Is this really justified given the different penetration depths (as alluded to elsewhere in the paper)?

**Response 2:**

The discussion in the paper regarding changing penetration depths with frequency is most significant for those parameters which impact the scattering properties of the snowpack (correlation length, density, and layer thickness), and generally see greater variability. Although we do not expect temperature to have a strong direct effect on emissivity, we highlight the non-isothermal nature of a snowpack when simulating emissivity, whereby it is possible to use Eq. 3 to avoid using an average snowpack temperature. For the purpose of the observation emissivity retrieval, it is necessary to use effective temperature calculated for the channels centred on gaseous absorption lines, as discussed in the paper (line 130).

With the channels available on MARSS and ISMAR we are able to use the published method of Harlow (2009) to retrieve the effective surface temperature at 118 and 183 GHz. Emissivities at 89, 157 and 183 GHz (MARSS) are then calculated using the 183 GHz temperature, and those at 118 and 243 GHz (ISMAR) using the 118 GHz temperature. Based on differences between the 118 and 183 GHz effective temperatures, we expect a maximum error in the effective temperature assumption to be of the order of ±3 K, similar to the average variability in temperature observed in the snow pits of 2.9 K. The corresponding error in emissivity is of the order of 0.01, which when added in quadrature to the observation error estimates, increases the emissivity uncertainty from 0.01 to 0.014 at 89 GHz, and from 0.02 to 0.022 at 243 GHz. However, we would not expect these small differences in emissivity error to have a significant impact on the SCEM-UA retrievals, and given the computational expense, we do not feel there would be significant benefit from re-running the retrievals to include these errors.

**Changes:**

- Sect. 2.2, line 143, the following text has been added: "*These errors do not include uncertainty due to frequency-dependent changes in Ts,eff , which would have the greatest impact at 89 GHz, increasing uncertainty by a maximum of 0.004. This corresponds to a maximum error in Ts,eff of ±3 K, based on differences between 118 and 183 GHz Ts,eff , and average variability in observed snowpack temperature of 2.9 K. This is not expected to have a significant impact on the SCEM-UA retrievals.*"
- The order of the two sentences preceding the above inclusion has been changed to improve the flow of the narrative.
- Line 131, additional detail has been added to describe the emissivity retrieval: "*By assuming that Ts,eff is approximately independent of frequency, Eq. (1) is used to calculate e at window channel frequencies. To ensure consistent viewing geometry, Ts,eff at 183 GHz is used to calculate e for the MARSS channels (89, 157 and 183 GHz), and Ts,eff at 118 GHz to calculate e for the ISMAR channels (118 and 243 GHz).*"

While addressing changes relating to this comment, the following changes were also made:

- Equation 3 has been updated to use *"Tbsmrt"* for simulated upwelling, rather than *"Tb"*, for consistency with the text.
- The text at line 125 of the revised manuscript has been amended to use the variable names *"Tb,up,surf"* and *"Tb,down,surf"* rather than *"surface upwelling and downwelling brightness temperatures"* as these variables have already been defined.

**Comment 3:**

L168-169/Table 1: Strictly, the table gives correlation length, thickness and density, rather than SSA, thickness, density and temperature. The conversion of SSA to correlation length is only introduced later in the paper which might be confusing to some readers. Correlation length is probably indeed the better quantity to show in the table, so I suggest introducing the conversion earlier.

**Response 3:**

We acknowledge there is a mismatch between the contents of Table 1 and the text at Line 168/169 in the original manuscript. This table was intended to summarise the parameter ranges used in the retrievals, rather than summarising all the snow pit observations.

In the revised manuscript we therefore suggested moving the table to Sect. 2.6.2, which discusses the use of parameter ranges of correlation length, layer thickness and density in the SCEM-UA retrievals, and occurs after the conversion of SSA to correlation length has been introduced.

**Changes:**

- Table 1 has been moved to Sect. 2.6.2, page 12, and become Table 2.

**Comment 4:**

Sections 2.2., 3: The use of cluster mean emissivity spectra (rather than individual spectra) seems an important choice which I feel should be motivated better, including a discussion of the implications. It has important implications on the results and their interpretation:

**Response 4:**

Firstly, this comment has highlighted that the text and Fig. 3 caption are incorrect, as the observed spectra used in the retrievals were cluster centroids, rather than cluster means.

**Changes:**

- Fig. 3 caption, and the text at line 150 in the revised manuscript, have been updated to refer to cluster centroids.

**Comment 4a:**

Presumably, it means that the RMSE & MAE shown in table 3 and elsewhere are based on 5 values (ie the 5 frequencies considered). It would be worth making this clear in the text – it's not a very large sample to calculate statistics from. Using individual spectra would increase the sample size and produce presumably more informative statistics.

**Response 4a:**

We agree that calculating the statistics, particularly RMSE, using only the five frequency values is a limited statistical analysis. As Table 3 was primarily intended to quantify the simulation-observation difference from Fig. 5, with the in-text discussion focussing on the overlap between the standard deviation of the simulated spectra and the observation error, we suggest that MAE is the more relevant value, and given the limitations of calculating the RMSE in this way, we propose removing RMSE from Table 3 altogether.

We discuss the suggestion of running the retrieval for individual spectra in response to part 2 of this comment (Response 4b).

**Changes:**

- RMSE has been removed from Table 3 and throughout the text.

**Comment 4b:**

The cluster means are necessarily smoother in frequency than some of the individual spectra shown in Fig. 3. I would expect that less structure makes it easier to find a parameter set that is able to model the spectra. So I suspect the RMSE values are a little optimistic as a result of using the cluster means as observations? An alternative would be to try and fit the individual spectra and calculate RMSEs from these results, still keeping the clustering to separate different regimes (with different demands on the modelling). This would increase the sample size and, provided RMSEs are similar, it would strengthen the finding that SMRT is capable of modelling very diverse spectra.

**Response 4b:**

As mentioned in the initial response to this comment, the observed spectra were cluster centroids rather than cluster means. As the cluster centroids are taken from an observed spectrum, they preserve the shape of that spectrum, as opposed to the smoothing that would result from a cluster mean. We hope this addresses concerns around how easily the retrieval was able to model the spectra.

Both parts of this comments suggest running the retrieval for individual spectra, however, the decision to cluster the observations and use cluster centres to represent observations, was made to reduce the computational cost of running the retrieval for thousands of spectra. We determined that the clustering method allowed us to represent the variability observed, while using the cluster centres preserves the different shapes of the observed spectra. Although we agree running the analysis for a greater number of observations would strengthen the statistics, we would not expect to see large differences in the retrieved parameters between similar observed spectra in a cluster group.

**Comment 5:**

Section 3: The snowpack parameter retrievals exhibit considerable differences at least for some clusters. I wonder whether these differences are backed up by the snowpack measurements. For instance, the snowpack measurements could be grouped into the 8 clusters based on the aircraft data that is most appropriate in terms of location and timing. Based on this grouping, is there a tendency for larger/smaller thickness or correlation length measurements in line with the retrieved values? I appreciate that the limited number of the pit-measurements and the heterogeneity may prevent such an analysis from being meaningful, but I would nevertheless be interested if this has been considered or attempted.

**Response 5:**

We agree that this is an interesting comparison to make, and in response to this comment we attempted an analysis where each pit was assigned a cluster according to its nearest aircraft observation. Some similarities could be seen between the snow pit parameters and those retrieved by SCEM-UA, such as wind slab thickness of pits near clusters 0 and 3 being shallower than those near clusters 5 and 7, as was also seen in the retrieved parameters. However, as suggested in the comment, the limited number of pits meant that not all clusters were represented by the pit locations, and some clusters were only represented by 1 or 2 pits. It is therefore not possible to draw any meaningful comparisons using the pit data.

References

Harlow, R. C.: Millimeter microwave emissivities and effective temperatures of snow-covered surfaces: Evidence for lambertian surface scattering, IEEE. T. Geosci. Remote., 47, 1957–1970, https://doi.org/10.1109/TGRS.2008.2011893, 2009

Sandells, M., Rutter, N., Wivell, K., Essery, R., Fox, S., Harlow, C., Picard, G., Roy, A., Royer, A., and Toose, P.: Simulation of Arctic snow microwave emission in surface-sensitive atmosphere channels, EGUsphere [preprint], https://doi.org/10.5194/egusphere-2023-696, 2023

**Response to Referee #2**

General

The paper presents an analysis of snow emissivity simulations using SMRT at high microwave frequencies, using a dataset collected in the Canadian Arctic to drive the model. Airborne data collected at the relevant frequencies are used to derive statistical ranges for spectral behaviour of emissivity, and these are compared to emissivity iterations generated using a coupled system of SMRT with SCEM-UA. Posterior distributions of simulation parameters most affecting emissivity (correlation length, layer thickness and density) are analysed in depth.

The paper is well structured, and the writing is excellent, making it a very pleasant reading experience. The introduction provides a nice overview of both the potential and challenges related to applying high frequency microwave observations to support NWP. The applied datasets are described very concisely, but this can be justified as the data are already described in previous works (which are correctly referenced). The results indeed indicate that SMRT is applicable well beyond 100 GHz for simulating snow emissivity, given the correct input information – this by itself is a very encouraging step considering the use of observations at these frequencies over snow covered areas. The importance of snow microstructure and layer thickness are highlighted as the most relevant parameters. The remaining challenges in acquiring such snow information, e.g. using Land surface Models, are also outlined in the conclusions. With minor modifications (see comments), Figures and Tables are generally clear and well-prepared.

To conclude, I struggled to find any points to criticise even after several read-throughs. Therefore, I suggest to accept the paper taking into consideration the few minor comments below.

Minor comments

**Comment 1:**

Title: I find the title does not fully reflect the point of your paper, which was to evaluate which factors affect the capability of SMRT to simulate emissivity of snow at high frequencies for purposes of NWP schemes (this is at least my interpretation. Something like "Potential of SMRT to simulate emissivity of Arctic tundra snow at 89 to 234 GHz for Numerical Weather Prediction" would maybe grasp the interest of more readers?

**Response 1:**

We agree that the title could be more descriptive, and should include the frequency range used, given this is higher than frequencies typically studied. However, while this paper demonstrates that SMRT is capable of simulating emissivity at these frequencies given appropriate inputs, there are still other steps that need to be explored further for SMRT to be utilised in an NWP system, such as a suitable snowpack model, as discussed in the paper. Therefore, we feel it would be misleading to suggest in the title that this alone demonstrates SMRT's potential for use in NWP.

**Changes:**

-   The title has been updated: *"Evaluating Snow Microwave Radiative Transfer (SMRT) model emissivities with 89 to 243 GHz observations of Arctic tundra snow"*.

**Comment 2:**

Lines 134-142. As I understand it, the observation error values of 0.01 and 0.02 you have used are only derived from the consideration of imperfections in assumed atmospheric conditions. What about instrumental uncertainties? What is the actual observation uncertainty (in TB) of MARSS and ISMAR, and how does this translate to observation uncertainty of emissivity? I suspect that the effect is small, but it could be mentioned somewhere.

**Response 2:**

The observation errors of 0.01 and 0.02 do also include an estimate of instrument uncertainty of ±1 K, however we agree this could be made clearer in the text.

**Changes:**

-   Line 138, the text was updated to include the instrument uncertainty estimate: *"An evaluation of the sensitivity of emissivity retrievals to errors in observed brightness temperatures (of ±1 K), and assumed parameters, such as temperature and humidity of the atmospheric layer between the aircraft and the surface, suggests that retrieval error is approximately 0.01 for frequencies between 89 and 157 GHz, and 0.02 for frequencies between 183 and 243 GHz."*

**Comment 3:**

Table 1: give meaning of SS, WS and DH in the text (lines 168-169) and in the Table caption. Same applies to Table 2. Now, these acronyms are finally spelled out in caption of Table 4. Probably fine to leave it there also.

**Changes:**

-   As suggested, surface snow (SS), wind slab (WS) and depth hoar (DH) are now defined in the text at line 174 of the revised manuscript, and in the captions of Table 1 and Table 2.

**Comment 4:**

Line 211. Add a note in the text somewhere that in 2-layer simulations, it was the surface snow layer that was omitted. This is of course apparent from Table 2, but you could add the mention just for sake of clarity.

**Response 4:**

The difference between the two- and three-layer snowpacks is discussed in more detail in Sect. 2.6.1, line 238 (line 243 in revised manuscript), however, we agree that the difference could be made clearer when first introducing the retrievals in Sect. 2.6.

**Changes:**

- Line 215 has been updated in the revised manuscript: *"The experiments were also split for two- and three-layer snowpacks, whereby two-layer snowpacks did not have a surface snow layer, to reflect the different stratigraphy observed during MACSSIMIZE".*

**Comment 5:**

Table 3. Consider presenting RMSE and MAE as relative values (%) to the observed emissivity. To me at least, it would make the numbers easier to grasp, and would make the table prettier, reducing the amount of decimals. Perhaps this applies also to the text (including abstract); saying that "RMSE decreased from 0.029 to 0.013, and MAE from 0.018 to 0.0078" says very little to a person not familiar with emissivity errors.

**Response 5:**

Please see response to Referee #1, comment 4, regarding RMSE. Given the limitations calculating RMSE with 5 values, we suggested removing RMSE from the discussion and focussing on MAE to quantify the simulation-observation differences in Fig. 5. We feel including the absolute value of MAE is useful for the comparison, however, agree that including a % of the observed emissivity will also be useful.

**Changes:**

- Table 3 has been updated to include Mean Absolute Percentage Error (MAPE).
- All reference to RMSE and MAE throughout the manuscript have been updated to reflect the removal of RMSE, and addition of MAPE. Lines 10-11, 269, 280, 285, 289, 371.